# Therapeutic Potential of Mesenchymal Stromal Cells and Extracellular Vesicles in the Treatment of Radiation Lesions—A Review

**DOI:** 10.3390/cells10020427

**Published:** 2021-02-18

**Authors:** Mohi Rezvani

**Affiliations:** 1CSO, Swiss Bioscience GmbH, Wagistrasse 27 A, Schlieren, 8952 Zurich, Switzerland; mohi@swissbioscience.com; 2Reader Emeritus, Faculty of Clinical Medicine, University of Oxford, Kidlington, Oxfordshire OX5 2AX, UK

**Keywords:** radiation, mesenchymal, stem cell, extracellular vesicles, micro vesicles, paracrine effect, adipose tissue derived stem cells

## Abstract

Ionising radiation-induced normal tissue damage is a major concern in clinic and public health. It is the most limiting factor in radiotherapy treatment of malignant diseases. It can also cause a serious harm to populations exposed to accidental radiation exposure or nuclear warfare. With regard to the clinical use of radiation, there has been a number of modalities used in the field of radiotherapy. These includes physical modalities such modified collimators or fractionation schedules in radiotherapy. In addition, there are a number of pharmacological agents such as essential fatty acids, vasoactive drugs, enzyme inhibitors, antioxidants, and growth factors for the prevention or treatment of radiation lesions in general. However, at present, there is no standard procedure for the treatment of radiation-induced normal tissue lesions. Stem cells and their role in tissue regeneration have been known to biologists, in particular to radiobiologists, for many years. It was only recently that the potential of stem cells was studied in the treatment of radiation lesions. Stem cells, immediately after their successful isolation from a variety of animal and human tissues, demonstrated their likely application in the treatment of various diseases. This paper describes the types and origin of stem cells, their characteristics, current research, and reviews their potential in the treatment and regeneration of radiation induced normal tissue lesions. Adult stem cells, among those mesenchymal stem cells (MSCs), are the most extensively studied of stem cells. This review focuses on the effects of MSCs in the treatment of radiation lesions.

## 1. Development of Radiation-Induced Lesions

It is now well accepted that the human body contains adult stem cells or in other words post-natal stem cells that are capable of differentiating into other tissues and can regenerate or repair damaged tissues. Over the last decades, stem cell hypothesis, the development of tissue deficits due to the inability of stem cells to replenish lost cells, has become a reality. Stem cells were in a way studied by radiobiologists well before it was proposed as a hypothesis. In fact, the initial theory of the development of radiation lesions’ “target cell theory” was based on radiation-induced cell loss. Target cell theory introduced by Puck and Marcus [1] considers cell loss as the cardinal cause of radiation induced normal tissue damage or tumour ablation. In recent years, it has been shown that the process of development of radiation damage and the damage itself starts by molecular changes long before denudation of target cells. However, one cannot deny the fact that the ultimate lesions manifest as loss of functional cells. Most bodily tissues possess a pool of clonogenic cells that are mobilised in response to assaults such as trauma or radiation. Damage to the tissue is repaired by proliferation of clonogenic or tissue specific stem cells. Sterilisation of these clonogenic cells by radiation manifests as radiation damage. In mild cases as the damage is sensed, these clonogenic cells migrate to the site of damage, and together with local surviving clonogic cells, proliferate to repair the tissue. However, in severe cases of tissue repairs, there might not be enough surviving clonogenic cells as the site of damage or sufficient number of mobilised cells to reach the site and repair the damage. Thus, the damage gets established as a result of failure of endogenous stem cells to regenerate the damaged tissue.

In early responding tissues, such as gut, oral mucosa, or epidermis of skin, the initiation of molecular events triggered by radiation results in the loss of both clonogenic and differentiated functional cells. Loss of clonogenic cells or in other words basal stem cells results in a deficiency to replace the lost functional cells. In the event of survival of a sufficient number of proliferating tissue specific stem cells in the irradiated region or its vicinity, complete healing is observed. However, in severe cases, where the radiation causes sterilisation of the tissue-specific clonogeic cells, denudation of the tissue will follow. Deficiency of the stem cells to produce new cells to replace the lost cells and resulting imbalance brings about the erosion of the epithelial layer.

In late responding tissues, such as late dermal reaction of skin or central nervous system damage, the involvement of stem cells are also established. However, the pattern of development of lesions in late responding tissues is more complex, as the response of slow turnover tissues (such as neural tissue) differ from the response of rapid turnover tissues (such as epithelial tissue). In late responding tissues, the overall tissue response is dependent on more than one cell type and their response to irradiation. The complex process of late radiation damage is initiated by a cascade of molecular events from injured cells that result in eventual denudation of functional differentiated cells. The response develops as the consequence of damage to both slow and rapid turnover tissues. For example, a rapid unset of radiation-induced apoptosis has been reported as early as 3–6 h in dentate gyrus after irradiation of rat brain [2,3]. These authors also reported a higher number of apoptosis than the number of proliferating cells and concluded that non-proliferating cells as well as proliferating cells in the subgranular zone of rat brain were sensitive to radiation and cell number, in this region, was significantly lower than age-matched controls 120 days after irradiation. This in part can be the cause of radiation-induced cognitive deficit. A dose-dependent reduction in the number of subepidermal cells in irradiated rat brain and the inability of surviving stem cells in regenerating the subepiderma, that manifest a clear deficit at 180 days after irradiation, was reported [2]. Deficiency of stem cells to regenerate the lost tissue results in the development of scaring or fibrosis as a final lesion. Therefore, the replacement of stem cells by donor stem cells, possibly before establishment of the lesion, may prevent the development or shorten the duration/severity of the lesions in both early and late responding tissues.

## 2. Treatment of Radiation Lesions with Stem Cells

Radiation lesions is amenable to treatment by methods that result in repairing or regeneration of the damaged tissue. In fact, stem cell transplantation in medical practice is not new and have been used for decades in bone marrow transplantation [4].

Stem cell treatment of radiation damage is based on the assumption that the transplanted cells integrate with the damaged host tissue to replace the damaged/lost cells or stimulate the host cells to prevent the damage or regenerate the damaged tissue. The later will obviously be more efficient before establishment of the radiation damage. Transplanting the stem cells before the full establishment of radiation lesion can prevent the development of radiation damage or shorten the duration of the manifestation of the lesion.

Bone marrow transplantation has been successfully used in the treatment of leukaemia, lymphoma, and certain types of anaemia procedures. Initial efforts in this field were directed towards transplantation of pre-differentiated stem cells and a good example of this is bone marrow transplantation that started as early as 1951 with the work of Lorenz [5] who found that infusion of the spleen or marrow cells could protect the irradiated mice. Bone marrow transplantation is based on allogenic use of stem cells. Whole marrow or stem cells of the marrow are extracted from a donor and transplanted to the host to reconstruct the haemopoietic tissues of cancer patients. The patient, prior to bone-marrow transplantation, is myeloablated by radiation or chemotherapy. The process of bone marrow transplantation is reviewed by [4].

Later, non-tissue specific or naive stem cells were transplanted on the basis of the opinion that the niche, or local microenvironment, consisting of surrounding cells, will define the fate of the transplanted cells and direct the administered stem cells to lodge into target tissue and differentiate into the required cells to restore structural and functional deficits.

In this article, a number of papers indicating the application of stem cells in the treatment of radiation-induced lesions are reviewed. It is also argued that the beneficial effect of transplanted stem cells in irradiated bodies is not necessarily due to the lodging of the transplanted stem cells in the irradiated tissue to replace the lost/damaged cells. It is suggested that perhaps the result is by paracrine effect; i.e., transplanted stem cells secrete bioactive substances that are capable of stimulating the host cells to reproduce and repair the damaged tissue. This means that the transplanted stem cells, besides integrating in the structure of damaged tissues, secrete biologically active factors, mainly in the form of extracellular vesicles, such as exosomes and microvesicles, that stimulate and mobilise the endogenous stem cells to repair the damage. Recently, it was shown by many researchers including ourselves, that the effect of stem cells is exerted in a paracrine fashion [6,7,8]. Transplanted stem cells, by integration with the host tissue, mobilisation of endogenous stem cells, or a combination of both mechanisms, result in functional and structural improvements of injured tissues. For a review on extracellular vesicles, see [9,10].

## 3. Types of Stem Cells

Stem cells are undifferentiated cells that are capable of dividing to produce more stem cells and/or differentiate specialised cells. Stem cells are classified by their potentiality into three main types; multipotent, pluripotent, and totipotent. Totipotent stem cells can generate an entire individual. Pluripotency is the ability of certain cells to differentiate into the three embryonic layers (ectoderm, mesoderm, and endoderm). Multipotency is the ability of stem cells to differentiate into one or two embryonic layers such as mesoderm and endoderm. In contrast, adult stem cells are multipotent cells.The stem cells currently used in medical applications or studied in research can be divided into three main types.

(1)Embryonic stem cells (ES): these are pluripotent cells located at the inner cell mass of blastocysts. Embryonic stem cells are usually harvested around four days after fertilisation when the embryo is in its blastula phase [11]. Embryonic stem cells can be differentiated into any one of the three germ layers; endoderm, mesoderm, or ectoderm.(2)Induced pluripotent stem cells (iPCs): these cells, as indicated by their name are pluripotent that are generated from mature somatic cells, like skin or blood cells, by introduction of transcription factors for encoding certain genes. This is in fact back reprogramming of mature cells to embryonic stem cell state. The classic mixture of transcriptions factors to produce iPSCs consist of Oct3/4, Sox2, Klf4, and c-Myc [12].(3)Adult stem cells: This is another group of stem cells that are multipotent. Adult stem cells or adult progenitor cells are tissue-specific stem cells are available almost in all body tissues [13] such as epidermal stem cells of skin, stem cells of human hair follicles, cardiac stem cells of heart, neural stem cells of the brain, hepatic stem cells, intestinal stem cells, dental pulp stem cells, ovarian epithelial stem cells, mammary stem cells, testicular stem cells, and satellite cells/myogenic stem cells of the skeletal muscle. Hemopoietic stem cells and mesenchymal stem cells are other groups of adult stem cells. Hemopoietic stem cells are derived from blood vessels and bone marrow. Mesenchymal stromal cells (MSCs) are another type of multipoint adult cells [14,15,16] found in bone marrow, adipose tissue [17,18], and almost all postnatal tissues [19]. MSCs are non-hematopoietic stem cell-like cells first identified by Friedenstein [20,21] and their characteristics are described [22]. In bone marrow, MSCs have a supportive role for hematopoietic stem progenitor cells (HSPCs) that is also involved in the maintenance of marrow microenvironment by secreting bioactive factors [23]. MSCs of adipose tissue are termed Adipose Tissue-derived Stem cells (ADSCs), which, like other MSCs are spindle-shaped plastic adherent cells, capable of differentiating to other cells [24,25]. Another source of MSCs (UC-MSCs) is umbilical cord blood [26] or Wharton jelly of umbilical cord [27,28]. UC-MSCs like other MSCs differentiate into three germ layers and contribute to tissue repair and regeneration [29].

ES and IPS cells have the advantage of indefinite renewal and the ability to differentiate into all cell types. This property gives them a role in replacing damaged cells by direct differentiation. On the other hand, adult stem cells are limited in their proliferation. Adult stem cells can either differentiate to replace specialized cells but in a limited number of cases. This is the case, for example, with MSCs that differentiate into osteoblasts. On the other hand, when adult stem cells come to repairing tissue from which they did not originate, they preferentially act by trophic effect, such as MSCs to allow intestinal regeneration.

The Mesenchymal and Tissue Stem Cell Committee of the International Society for Cellular Therapy [30] states three conditions as the minimal criteria for definition of human MSC. (1) MSC must be plastic-adherent, (2) express CD105, CD73, and CD90, and lack expression of CD45, CD34, CD14 or CD11b, CD79alpha or CD19 and HLA-DR surface molecules, and (3) differentiate into osteoblasts, adipocytes, and chondroblasts in vitro. MSCs has been shown to differentiate into endodermal lineage such as hepatocytes [31], cardiomyocytes [32], and ectodermal lineage neurons [33].

MSCs are the most extensively studied adult stem cells and BM-MSCs are the first to be transplanted and used in regenerative medicine, including treatment of radiation lesions. Alternatively, ADSCs appear to be a better kind of MSCs [34]. Furthermore, ADSCs can be obtained by lipoaspiration, which is much less invasive than obtaining BM-MSCs by bone marrow aspiration. ADSCs exhibit intermediate radiation sensitivity [35] and it appears that irradiation of human ADSCs with low-level laser changes their morphology and enhances their proliferation and therapeutic potential [36]. The potential of mesenchymal stem cell therapy in the treatment of radiation-induced lesions has been reviewed [37].

## 4. Homing of Transplanted Stem cells

MSCs, for regenerative purposes, can be transplanted directly into the site of damage or introduced systemically. In the latter, it is assumed that homing of the transplanted cells is regulated by the local microenvironment and they are directed to the site of injury by cues from damaged tissues of the host through a series of signals. Furthermore, the transplanted cells secrete diverse trophic factors and immunomodulatory substances that contribute to the process of regeneration by stimulating the endogenic stem cells. In majority of the studies of the distribution of transplanted cells in irradiated animals, it has been shown that the transplanted cells home to the radiation-damaged tissues. MSCs intravenously transplanted to rats with myocardial lesions home to the infarct region of the heart, while in uninjured control animals, the transplanted cells migrated to the bone marrow [38]. In the treatment of radiation-induced multi-organ failure in non-human primates, transplanted MSCs home to injured tissues [39]. Human MSCs were systemically transplanted into total body or abdominal irradiated NOD/SCID mice [40,41]. It was reported that the transplanted cells home to the irradiated organs and were found three months post irradiation. These observations support the hypothesis that transplanted stem cells migrate to radiation-induced injury sites in irradiated animals. However, this does not seem to be specific to radiation lesions as migration of transplanted stem cells to non-radiation damaged tissues has been reported too. In an acute nontransmural myocardial infarct model [42], it was shown that transplanted MSCs mainly home to the infarct myocardial region observed 24 h after intravenous transplantation that lasted for 7 days after transplantation. However, these authors observed some migration to non-target organs as well but the main concentration was in the infarct region.

Homing factors are crucial in the delivery of stem cells to damaged tissues. Some homing factors have been identified. For example stromal cell-derived factor-1 (SDF-1) is known to allow the targeting of hematopoietic stem cells to the marrow when it needs to be recolonized by hematopoietic stem cells. The secretion of SDF-1 similarly allows the homing of MSCs that express the C-X-C Motif Chemochine Receptor-4 (CXCR4) molecule, which is the receptor for the SDF-1 molecule. Another chemokine, Monocyte Chemotactic Protein-1 (MCP-1), was found to be a key regulator for stem cell recruitment to the myocardium in or cochlear tissue.

## 5. Stem Cell Treatment of Radiation Lesions

Interest in the application of mesenchymal cells as therapeutics has increased recently. A few early stage clinical trials have also been reported [43,44,45,46] but in general one can say that treatment with MSCs is still in an experimental phase and larger clinical trials are needed before its clinical use. Safety of MSCs in clinical trials have been reviewed and adverse effects listed [47]. The safety of MSCs for the treatment of radiation lesions has also been reported [48].

Like other cells, irradiation of MSCs induces senescence and/or apoptosis [49]. This has been shown in MSCs isolated from irradiated human skin, where colony formation, proliferation, and differentiation capacity are reduced [50].

MSCs have been shown not to give rise to tumours [51] as they are non-tumourgenic [52].

## 6. Studies on Hematopoietic System

Although interest in stem cell treatment increased over the last two decades, stem cell transplantation started more than half a century ago with bone marrow transplantation by Lorenz et al. [5] followed by Barnes et al. [53]. These authors demonstrated that transplantation of bone marrow cells could protect mice against ionising radiation. This was the pioneering process of bone marrow transplantation that developed as a routine clinical procedure, where whole marrow or marrow cells extracted from bone marrow are transplanted into myeloablated host in the treatment of both malignant and non-malignant diseases such as leukaemia, lymphoma, and certain types of anaemia [54].

The effect of transplantation of bone marrow-derived mononuclear cells in non-human primates were studied by Bertho et al. [55]. These authors demonstrated that cell transplantation 24 h after 8 Gy total body irradiation shortened the period and severity of pancytopenia. Acute radiation syndrome (ARS), besides multi-organ failure, causes pancytopenia too. The efficacy of transplantation of human UC-MSCs to combat the effects of ARS was also studied [56]. However, in this study, UC-MSCs were modified to to express human extracellular superoxide dismutase. The regenerative potential of MSCs combined with the antioxidant effect of human extracellular superoxide dismutase was intended to produce a rapid and effective strategy for the treatment of radiation accident victims.

The protective effects of allogenic stem cell transplantation against acute radiation syndrome was demonstrated by transplantation of human umbilical cord-derived MSCs in mice [57].

## 7. Studies on Nervous System

Study of the regeneration of nervous system after irradiation was first started by transplantation of oligodendrocyte progenitor cells [58,59]. The results showed significant remyelation of radiation-induced demyelinated rat spinal cord.

Later, regenerative properties of transplantation of two types of neural stem cell were studied in a rat model of radiation myelopathy [60]. Twelve millimetre of rats’ spinal cord was irradiated with 22 Gy gamma rays. This was ED_100_ in six months in this model of radiation myelopathy. Neuroepithelial stem cells were obtained from the hipocampal proliferative analogue on embryonic day 14 from an H-2kb-tsA58 transgenic mouse. It was believed that both cell types were multipoint stem cells because they were 90% nestin positive in culture and they had been shown to differentiate into neurons, oligodendrocytes, and astrocytes. Stem cells were transplanted, intradurally, three months after spinal cord irradiation. While control animals developed front leg paralysis within 183 days after irradiation, 30% of animals in stem cell transplanted group stayed paralysis free until day 200.

Wei et al. [61] using rat cervical spinal cord irradiation model irradiated 20 mm of cervical spinal cord of rats and injected one million UC-MSCs through the tail vein at 90 days after injection followed by three weekly injections. These authors demonstrated that multiple injections of stem cells significantly improved neuron survival and locomotor recovery at 180 days post irradiation.

In a rat model of cranial irradiation [62], human embryonic stem cells were transplanted to the hippocampus of athymic nude rats two days after 10 Gy cranial irradiation. This resulted in a significant cognitive improvement four months after irradiation compared to the controls that did not receive stem cell transplantation. The same authors observed similar results in the same model after transplantation of human neural stem cells [63]. These authors reported equivalent cognitive restoration with both types of stem cell transplantations [64]. Efficacy of stem cell therapy in amelioration of radiation-induced brain damage is reviewed by Chu et al. [65].

## 8. Studies on the Gut

Semont et al. [66] studied the regenerative effects of transplantation of human BM-MSCs in NOD/SCID mice. Transplantation was by infusion and the results were assessed by functional and histological assessment of the jejunum. The results demonstrated both structural and functional improvements by MSC transplantation.

The effect of autologous bone marrow derived stem cell treatment was studied in a pig model of irradiation proctitis, developed by 4MV photons [67]. It was demonstrated that repeated administration of mesenchymal stem cells resulted in reduction of collagen deposition and radiation-induced fibrosis. Reduction in expression of inflammatory cytokines both systemically and in rectal mucosa were also observed.

In a rat model of colorectal cancer, transplantation of allogenic MSCs significantly improved normal tissue damage induced by radiotherapy [68]. This study also demonstrated that MSC transplantation increased the tumour-free survival of the animals. The number of tumour free animals was higher than expected while the incidence and size of the tumours were reduced.

In our own laboratory (unpublished work), the effect of transplantation by ip injection of human ADSCs on gut was studied in rats. In this study, four cm of rats’ distal colon were irradiated with 11 Gy 250 kV X-rays while the rest of the animal was shielded. Twenty four hours after irradiation, the animals were grouped into six groups and treated. Group 1: unirradiated controls received only one ml PBS injection, Group 2–6 received radiation followed by one ml saline injection (radiation only- Group 2), two million ADSCs suspended in one ml PBS (Group 3), two million ADSCs lysate in one ml saline (Group 4). One ml conditioned media collected from 2 million ADSC cultures (Group 5) injected ip and finally conditioned media administered three times 24 h, 72 h, and 120 hrs after irradiation (Group 6). The results were assessed by counting the number of crypts per circumference by light microscopy nine days after irradiation. As expected, radiation only reduced the number of crypts significantly compared with unirradiated control group. Injection of 2 million intact ADSCs, lysate, or a single dose of conditioned media increased the number of crypts almost equally. However, the best result was obtained by three consecutive injections of conditioned media. Comparable results obtained from injection of intact MSCs or the lysate of the equivalent number of cell indicates the possibility of a paracrine effect. This was also confirmed that the outcome of conditioned media injection that usually contains mi-RNA, a number of proteins, and biologically active lipids was more effective than the intact stem cells injections.

The possibility of the paracrine effect was indicated in a similar study [69] where the effectiveness of secretions of human UC-MSC to prevent radiation-induced intestinal injury was investigated in BALB/C mice after 10 Gy cobalt irradiation. In this study, UC-MSCs were expanded under hypoxic conditions. Multiple injections of the hypoxic conditioned media was delivered to the animals after irradiation for seven days. This treatment improved the structure of the intestine, decreased diarrhoea, and increased the survival rate.

Paracrine effect of stem cell transplantation was also shown in a study by Chen et al. [70] where conditioned media obtained from rat bone marrow MSCs were injected into rats just before irradiation. The results indicated that the conditioned media injection increased the expression of anti-inflammatory cytokines and reduced the expression of inflammatory cytokines.

In a recent study [71], total body irradiated mouse, at a dose of 7 Gy (^60^Co γ-rays), received intravenous injections of one million human placenta-derived stem cells for 10 days after irradiation and compared with another group of animals that received radiation only. Ten days after irradiation, radiation-induced small intestinal damage was compared with that of a control group. It was shown that stem cell transplantation significantly improved (*p* < 0.01) the outcome of radiation enteropathy or lethal radiation syndrome. It was also shown that stem cell transplantation exerted inhibitory actions on inflammatory cytokines and assisted the re-establishment of epithelial homeostasis.

In a rat model of colonic anastomosis performed by irradiation [72], it was shown that transplantation of rat ADSCs promoted anastomotic healing of the irradiated colon through enhanced vessel formation and reduced inflammation. In this study, the ADSC injections were delivered several times before and after the surgical procedure.

Sémont et al. [66,73] described the effects of MSCs as a consequence of their ability to improve the renewal capability of the small intestine epithelium. They also suggested that MSC treatment favours the re-establishment of cellular homeostasis by both increasing endogenous proliferation processes and inhibiting radiation-induce apoptosis of the small intestine epithelial cells.

MSC treatment decreased the interactions between mast cells and nerve fibers and reversed mechanical visceral hypersensitivity [74]. These authors suggest that the mechanism of effect is that the MSCs release cytokines and growth factors, such as interleukin (IL)-11, human hepatocyte growth factor, fibroblast growth factor-2, and insulin-like growth factors. Each of these factors have been described earlier as facilitating intestinal mucosa repair, either through enhancement of cell proliferation or inhibition of epithelial cell apoptosis [66,69,73,74].

## 9. Studies on the Liver

Prevention of radiation-induced liver damage was the subject of study well before the establishment of mesenchymal cells as stem cells. In an earlier work [75], lethally irradiated mice were treated with syngeneic fetal liver cells that resulted in longer survival.

Later, the effects of BM-MSC transplantation on irradiated liver was studied in NOD/SCID mice [76]. In this study, animals received 10.5 Gy of ^60^Co gamma rays, followed by intravenous delivery of 5 million human BM-MSCs five hours after irradiation. This study demonstrated that MSC transplantation reduced radiation-induced apoptosis and significantly reduced the transaminase values (AST and ALT) compared with irradiated but not transplanted animals.

In a study of the effects of hepatic irradiation on transplanted BM-MSCs in cirrhotic rats and the underlying mechanism by which mesenchymal stem cells (MSCs) relieve liver fibrosis [77], the BM-MSCs from male rats were injected via portal vein into two groups of thioacetamide-induced cirrhotic rats. The right hemiliver of one cirrhotic rat group was irradiated (15 Gy) four days before transplantation. It was shown that the transplantation of MSCs alleviated liver fibrosis and reduced expression of transforming growth factor-β1, Smad2, and collagen type I. In addition, hepatic irradiation promoted homing and repopulation of MSCs and enhanced the effect of BM-MSCs in improving thioacetamide-induced liver fibrosis in rats. The authors concluded that BM-MSCs may function by inhibiting transforming growth factor-β-Smad signaling pathway in the liver.

## 10. Studies on the Lung

Mice exposed to thoracic irradiation were injected intravenously on days 0 and 14 after irradiation with genetically modified MSCs, expressing soluble transforming growth factor-b, MSCs conditioned media (MSC-CM). Sixty weeks after irradiation, all animals in the control group that had received only PBS injection after irradiation died. The survival rate of MSC and MSC-CM groups was 40% and 80%, respectively. The thickness of alveolar septa, malondialdehyde in lung homogenates, and plasma TGF-β1 levels significantly decreased in mice treated with either MSCs or MSC-CM, indicating the protective effects of MSC transplantation or MSC-CM injection, which reflects the paracrine effect of MSCs [78].

Improvements in acute radiation-induced lung injury has been demonstrated by Jiang et al. [79]. These authors injected rat ADSCs through the tail vein to right lung irradiated rats two hours after irradiation with 15 Gy X-rays. ADSC transplantation resulted in increased serum levels of anti-inflammatory cytokine IL-10 and reduced serum levels of the pro-inflammatory cytokines TNF-alpha, IL-1, and IL-6.

Human umbilical cord stem cells were transplanted 24 h before or 24 h after lung irradiation in rats [80]. The results demonstrated alleviation of radiation pneumonitis in both groups in comparison with the controls. Transplantation of umbilical cord MSCs have also been shown to be beneficial in the prevention of radiation-induced lung fibrosis [81,82]. However, these authors have shown that modification of stem cells to produce manganese superoxide dismutase significantly enhances the modulatory effect of MSC transplantation. Furthermore, MSC transplantation has been shown to reduce the incidence of lung metastasis in mice [83], beside lowering radiation-induced lung injury.

Feasibility and mode of action of mesenchymal stem cell therapy in amelioration of radiation-induced lung injury have been reported by Xu [84].

## 11. Studies on the Skin

Francois et al. [85] irradiated the skin of the hind leg of NOD/SCID mouse with 30 Gy single dose of Cobalt-60 gamma rays. Human BM-MSCs was transplanted by intravenous injection 24 h after irradiation. In stem cell transplanted animals, partial healing of the skin lesions was observed two weeks earlier; at six weeks after irradiation. Complete healing of epithelium was observed at eight weeks after irradiation in this group. While in control animals that had received radiation only, only partial healing of the skin lesions were observed at eight weeks.

BM-MSCs were injected into the skin of mini-pigs irradiated with large dose of 50 Gy of ^60^Co gamma rays [86]. Autologous BM-MSCs were injected intradermally 4–14 weeks after irradiation, 2–3 times a week. Each injection contained 99–128 million autologous cells. Minipigs were followed up for over 30 weeks and it was shown that the treatment lead to local accumulation of lymphocytes at the dermis/subcutis border, improved vascularization, and reduction of inflammatory reactions. In another study of acute cutaneous radiation syndrome [87], skin of mini-pigs were irradiated with 50 Gy of ^60^Co gamma rays. At day 76 post irradiation, inflammatory cytokines IL-1α and IL-6 (specific markers of M1 macrophage) and IL-10 and TGF-β (specific markers of M2 macrophages) were assessed. Treatment with autologous ADSCs resulted in increased M2 macrophage markers associated with CD68+/CD206+ cells, indicating that MSC treatment directed the inflammatory response to proregenerapive outcome.

ADSC treatment of irradiated wounds on rats resulted in accelerated healing of wounds in rats [88]. Three-cm diameter of rats dorsal skin was irradiated with 50 Gy of 6MeV electrons. Three weeks after irradiation, rats received one million MSCs in PBS, compared with those that received only PBS. At week six after irradiation, wounds on ADSC treated rats were significantly smaller than controls. Histological examination of the wounds also indicated re-epithelialsation and neoangiogenesis in MSC-treated wounds.

This was supported by the reported beneficiary effects of ADSC injection on healing of irradiated wounds in nude mice [89]. The dorsal skin of nude mice were irradiated non-lethally and wounds were created by skin biopsy punch. Wounds were injected with ADSCs and compared with vehicle injected wounds.

Beneficial effects of cell therapy was demonstrated after transplantation of bone marrow mononuclear cells (BMMNC) on irradiated wounds [90]. Skin wounds were created by skin biopsy punch after cobalt-60 irradiation. It was demonstrated that cell therapy resulted in increased vascular density and improved matrix remodelling.

With respect to the clinical effects of MSC transplantation on treating radiation-induced lesions, a 27-year-old Chilean radiation accident victim was treated by skin allograft after excision of the irradiated tissue [91]. The graft did not last very long and got infected. The patient was treated with skin allograft again but this time with addition of autologous BM-MSCs. A second dose of stem cells was delivered nine days after that, resulting in complete healing and wound closure at 75 days after first MSC transplantation.

A number of clinical studies that are not stem cell transplantation per se but can be attributed to the existence of ADSCs in fat have been reported. These include the treatment of radiation induced normal tissue lesions by autologous fat grafting. A 67-year-old cancer patient who developed a chronic non-healing ulcer in her leg after surgery and radiotherapy of a squamous cell carcinoma was treated with fat infiltrated around and under the ulcer area. The ulcer fully healed two months after treatment [92]. Rigoti et al. [93] treated 20 patients suffering from radiation-induced normal tissue lesions as side effects of radiotherapy with autologous fat grafting that resulted in improvements in all cases. Fat grafting was successfully used in rectifying aesthetic defects caused radiotherapy in head and neck cancer patients [94]. Breast irradiated patients do not respond favourably to allogenic reconstruction [95]. However, favourable outcomes and formation of new subcutaneous tissue have been reported after fat grafting in mastectomy patients who had received breast irradiation [96,97].

## 12. Studies on the Salivary Gland and Oral Mucosa

In a mouse model, the ability of ADSCs to minimize and/or repair single dose radiation-induced oral mucositis was demonstrated after 18 Gy single-dose of orthovoltage X-ray [98]. It was shown that intraperitoneal transplantation of 5 doses of 2.5 million freshly cultured syngenic ADSCs significantly and reproducibly reduced the duration of radiation-induced oral mucositis from 5.6 ± 0.3 days to 1.6 ± 0.3 days. The therapeutic benefits were shown to be significantly dependent on dose, frequency, and the start of cell transplantation.

Effects of BM-MSCs on irradiated salivary gland was assessed by mobilisation of autologus BM-MSC by administration of granulocyte stimulating factor (G-CSF) [99]. It was shown that the mobilised MSCs promoted regeneration of irradiated salivary glands and increased gland weight, number of ancinar cells, and salivary flow rate.

In another study [100], it was shown that the local transplantation of human ADSCs resulted in tissue remodelling with a greater number of salivary epithelial cells in a rat model of salivary gland irradiation. This indicated that local transplantation of ADSCs alleviated radiation-induced cell death. It was also shown that when an injectable porcine small intestinal submucosa matrix was used as a cell delivery carrier, the anti-apoptotic and anti-oxidative effects of ADSCs and salivary protein synthesis were enhanced.

Protective and regenerative effects of ADSCs on radiation induced salivary gland was also studied in rats [101]. These authors reported statistically significant improvements in the salivary gland of rats treated with ADMSc, 48 h after irradiation. The efficacy of stem cell transplantation and mobilisation in the treatment of radiation-induced xerostomia was discussed and reviewed [102,103].

Clinically, in a randomised placebo-controlled phase 1/2 trial [48], 30 patients were studied. In this study ADSCs or placebo were transplanted in submandibular glands of patients who had had previously received radiotherapy for oropharyngeal squamous cell carcinoma. No adverse events were detected from ADSC transplantation, indicating its safety. Unstimulated whole salivary flow rates in the transplanted group significantly increased compared to the placebo-arm. The xerostomia symptom scores significantly decreased and salivary gland function improved in the ADSC group.

## 13. Discussion

In a living body, cell loss and regeneration takes place continually as a natural process. Tissues function takes place as a result of a continued cell loss and replacement with new cells. Cells are lost due to ageing, wear and tear, or other insults such as radiation, and are replaced by new cells produced by indigenous stem cells or tissue-specific progenitor cells that differentiate into functional cells. Target cell theory of radiation damage [1] was developed exactly on this basis. According to this theory, cell loss is the cardinal cause of development of radiation lesions or tumour eradication by irradiation. In fact, radiation disturbs the usually continuous process of cell loss and cell replacement. The cells killed or damaged by radiation fail to produce sufficient progenies to replace the lost cells, therefore, the number of lost cells exceed the number of cells produced. When the deficit goes beyond a critical level where the number of progenies become so low that it cannot produce sufficient differentiated cells to maintain the tissue function, radiation lesion manifests. In early responding tissues, the latency period of the development of radiation lesion corresponds to the turnover time of the cells. For example, radiation mucositis, and radiation-induced moist desquamation of the skin are considered as a result of sterilisation of epithelia and their latency period corresponds to the turnover time of the target cells. However, late radiation damage cannot be described by turnover time of a certain cell type; however, it develops as a result of loss of a number of cells and subsequent events. For example, in the development of radiation-induced late dermal damage or late submucosal damage, loss or damage to endothelial cells play an important role. Loss of endothelial cells and damage to the vasculature impair the circulation and loss of parenchyma ensues. This is also true of radiation damage in two central nervous tissues, where late radiation damage manifests as demyelination of axons and necrosis. Some believe that the reproductive death of glial cells is the cardinal cause and demyelination and necrosis develop as a consequence of gradual loss of these cells [104]. However, some authors [105,106] consider vascular damage and lack of sufficient blood supply as the cardinal cause of the development of radiation-induced demyelination and necrosis of nervous tissue. Whatever the cause, both schools of thought agree that the demyelination and subsequent necrosis of nervous tissue is initiated by cell death, reproductive sterilisation of vascular or glial cells. The severity and duration of radiation-induced lesions are dose-dependent. This implies that the more cell loss, the more severe and long lasting the lesion. Besides radiation dose, radiation quality is another determinant factor on the degree of cell loss and consequently lesion development. However, treatment of radiation lesions, particularly treatment with stem cells, is in its infancy and there is not much data to be discussed.

Not all radiation lesions are fatal. Radiation lesions heal after sublethal doses; when surviving cells in the irradiated region regenerate or healthy cells from the margin of the irradiated region migrate to the irradiated area and revive the damaged tissue. However, when the cell loss is extensive or the number of surviving/migrating cells is not sufficient, the lesions remain unhealed. On this basis, replacement of lost cells by stem cell transplantation was a plausible attempt to modify radiation induced tissue damage.

Regeneration of irradiated salivary gland by mobilizing endogenous stem cells [99] supports the idea that there is always a number of stem cells in the damaged tissue and whole body, and their stimulation and mobilisation either by secretory factors from other stem cells or by cytokines could rescue damaged tissue. Protection of salivary glands from radiation-induced apoptosis and preservation of acinar structure and function were attributed to the activation of FGFR-PI3K signalling via actions of ADSC-secreted factors, including FGF10 [107].

The effectiveness of cell transplantation in amelioration of radiation lesions is supported by the works reviewed in this paper. Radiation lesions that develop due to lack or insufficiency of functional cells are modified by the transplantation of exogenous cells [90,108,109,110,111]. Besides, amelioration of radiation-induced lesions and subcutaneous tissue formation in patients who received fat grafting after mastectomy of breast-irradiated patients can be attributed to the stem cell component of the fat graft [96,97].

However, it is not certain that the beneficial effect of stem cell transplantation is the result of direct integration of transplanted cells in the damaged tissue or the result of stimulation of the surviving endogenous cells by the transplanted cells—the paracrine effect. Some authors, while reporting the beneficial effects of stem cell transplantation, fail to demonstrate the integration of the transplanted donor cells in host tissue or demonstrate a very low level of engraftment that cannot justify the significant functional improvements as a result of transplantation. In the study of the effect of stem cell transplantation on amelioration of radiation-induced salivary gland damage by mobilisation of endogenous bone marrow stem cells [99], significant improvements were seen in the gland weight and salivary flow but transdifferentiation of stimulated bone marrow cells in the salivary glands were not observed. Stem cell transplantation showed therapeutic effects on irradiated lung tissue but the number of transplanted cells in irradiated lungs were so low that they could not justify the observed improvements [78]. Neural stem cells transplanted intradurally in spinal cord irradiated rats resulted in 30% reduction in the development of radiation myelopathy [60] but these authors failed to demonstrate the transdifferentiation of the transplanted cells in the irradiated spinal cords of engrafted rats. Similarly, despite improvements in irradiated liver tissue by exogenous cell transplantation, the transplanted cells were not found in the liver of the irradiated animals [76].

These findings suggest that the beneficial effects of stem cell transplantation are not necessarily due to the replacement of damaged cells by healthy transplanted cells or their trans differentiation into functional cells. It is probable that the paracrine effect also plays a role [112,113,114]. It is to say that the transplanted cells secrete some bioactive factors that stimulate endogenous stem cells. Bioactive factors secreted by MSCs are both immunomodulatory and trophic. Secretion of angiogenic and antiapoptotic factors by transplanted human ADSCs have been reported [115]. VEGF secretion were increased manyfold when the ADSCs were cultured under hypoxic conditions. In fact, paracrine effect was reported as early as 1971 by Little [116] who reported the repair of potentially lethal radiation damage by a conditioned medium of cultured mammalian cells. Later, it was shown that the growth of cultured endothelial cells was enhanced and endothelial apoptosis was reduced by the addition of conditioned media obtained from ADSCs grown under hypoxic conditions [115]. Regeneration of radiation damaged tissues by transplanted MSCs has been attributed to the indirect effect of stem cell transplantation due to the secretion of cytokines and growth factors [76]. Tissue regeneration, acceleration of angiogenesis, and growth of nerves have been reported after transplantation of ADSCs in mice [117]. The beneficial effects of ADSC transplantation were attributed to the secretion of neurotrophic genes and extracellular matrix proteins required for nerve growth and myelination. MSCs, besides trophic effects, exert immunomodulatory effects too [109] that inhibit the surveillance ability of lymphocytes. This prevents the immunogenicity and allows allogenic transplantation of MSCs. A total of 73 proteins secreted by human ADSCs have been reported that includes factors such as heat shock proteins, macrophage inflammatory proteins, proteases, protease inhibitors, cycloskelethal components, extracellular matrix components, metabolic enzymes, anti-inflammatory proteinsVEGF, IGF-1, EGF, EGF, and many others [118,119]. Besides, RNA-containing microparticles are also involved in the paracrine effect. Microparticles or microvesicles consist of extracellular vesicles (EVs) that are released by almost all bodily cells, including stem cells. Evs are referred to a heterogenous population of membrane-coated small vesicles with diameter of 30–1000 nm. Exosomes constitute the microvesicles of diameter less than 200 nanometer. EVs consist of a bilipid membrane and a cargo consisting of various proteins and miRNA. Intracellular communication of cells is facilitated by secreted microvesicles [120,121,122,123]. Microvesicles released by stimulation of MSCs show therapeutic characteristics against ischemia-repurfusion induced acute and chronic kidney injury [124]. The same authors also demonstrated that inactivating RNA by pretreatment of microvesicles by RNase abrogated its therapeutic effect. This indicates the importance of the RNA component of microvesicles in exerting its therapeutic effect.

Evidence is mounting in support of paracrine effect of stem cells; in recent years particularly, EVs derived from stem cells have been the focus of extensive research efforts in the fields of regenerative medicine and radiation. The beneficial effects of MSC-secreted microvesicles have been demonstrated in vitro and in vivo treatment of many lesions [113,125,126,127,128,129,130,131,132,133]. Inhibition of tumour growth by MSC-derived microvesicles have also been demonstrated [68,128]. It has also been shown that platelet-derived microvesicles facilitate the homing of transplanted bone marrow stem cells in irradiated mice [134]. EVs extracted from human MSCs were injected into nude mice by three consecutive applications after a lethal whole body irradiation that resulted in 85% reduction in mortality [135]. Recently, the efficacy of MSC-derived EVs in amelioration of radiation-induced hematopoietic syndrome was reported [136]. Exosomes derived from mesenchymal stem cells have been used for conditioning macrophages to be used in the treatment of acute radiation syndrome [137]. It appears that, besides proteins and bioactive lipids, the RNA content of the cargo of EVs is the major component of the action of the beneficiary effects of EVs. This mode of action have been shown to be responsible for the amelioration of radiation-induced lung injury by mesenchymal cell-derived EVs [138]. The mode of action and potential of EVs in the treatment of radiation lesions are reviewed by Forsberg et al. [139]. EVs have also been indicated in mediating radiation-induced bystander effects [140].

## 14. Conclusions

The results of publications reviewed in this article indicate the beneficial effects of stem cell transplantation in the treatment of radiation lesions and tumour inhibition. Transplantation of intact stem cells or EVs derived from stem cells exert beneficial effects. However, it must be noted that radiation dose can play a major role in defining the results of the stem cell transplantation. The main principal of paracrine effect is based on the fact that paracrine factors excreted by transplanted stem cells stimulate endogenous stem cells to regenerate damaged tissue. After mild radiation doses, the donor cells partially contribute in regeneration of the damaged tissue and partially stimulate the endogenous stem cells to repair the damaged tissues. However, if a substantial large radiation dose is delivered to an organ, depleting almost all of the endogenous stem cells within the irradiated volume, the regeneration will be dependent almost entirely on the direct effect of the transplanted stem cells. This is to say that after a substantially large radiation dose, the paracrine effect will not be sufficiently effective and a substantially large stem cell dose will be required. Finally, it must be borne in mind that the conclusions made in this article are on the basis of limited experimental results published during recent years. Further research on the efficacy of stem cell transplantation and microvesicles secreted by activated stem cells, in amelioration of radiation lesions, is required.

## Data Availability

Not applicable.

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
