# Peer review of "Therapeutic Potential of Mesenchymal Stromal Cells and Extracellular Vesicles in the Treatment of Radiation Lesions—A Review"

_cells, 2021, doi:10.3390/cells10020427_

Round 1

Reviewer 1 Report

This article by Dr. Mohi Rezvani, entitled “Therapeutic Potentials of Mesenchymal Stromal Cells and Extracellular Vesicles in the Treatment of Radiation Lesions”, aims to review the current knowledge on the potentials of stem cell-based therapies in treating radiation induced damages. The topic is important and one that does not appear to have been reviewed often in a comprehensive manner. Dr. Rezvani has one review article in radiotherapy but less of a background in stem cells. This weakness is evident is aspects of his write-up. Overall, the main knowledge output of this review comprises of a collection of information on how infusion of MSCs/ADSCs/a few other stem cell types affect the recovery of various tissues following radiation therapies. Other than a tissue-based categorization, there is not much organization of the information or careful thought into what those reported effects might imply.

  1. Title and abstract: The title said “Mesenchymal stromal cells” and “extracellular vesicles”, and almost all references cited in this review looked at MSCs and/or ADSCs, with some using cord blood or placenta derived cells. Yet the author used the term stem cells quite loosely throughout (including abstract), which is misleading.
  2. Instead of trying to dig deep into the underlying events, the author presented the results mostly in a rather superficial, phenotypic way. There are many possibilities to explain the observed phenomenon, varying from tissue to tissue and among different stem cell types. Examples, a direct participation of MSCs into the regenerative process (which involves transdifferentiation), a protective role of MSCs, systemic immune response to infused MSCs, or a paracrine role on the endogenous cells, which may be stem or differentiated cells, to name a few.
  3. Neural tissue, which scarcely turns over, likely differs from epithelial tissues with constant turnovers in the response to radiation damage and MSC infusion.
  4. An issue regarding how systemically infused MSCs reach the targeted tissues (homing), especially if the presence of a barrier, such as BBB, is discussed in a general way but clearly all tissues are not equal.
  5.  P1, L28: The author needs to explain what is the “stem cell hypothesis”. There are potential several hypotheses.
  6. P1, L30: it is not clear what the author means by “other tissues”. The role of stem cells is to differentiate into cell types within the tissue. Their differentiation into other tissues (also known as trans-differentiation) does not normally occur biologically.
  7. It would be beneficial to include some diagrams.

Overall, I think the author fell short of digesting the knowledge and organizing/transforming it into an article that accurately depicts our current understanding of the field in sufficient depth and breadth.

Author Response

Reviewer 1:

  1. This comment is addressed in abstract by adding a paragraph to explain that MSCs were reviewed.
  2. I guess the reviewer has not paid due attention to the discussion section of this article.The points mentioned by the reviewer are discussed in detail the last three paragraphs of the discussion section of the paper.
  3. This comment agrees very well with the authors view but perhaps it was not very clear. therefore, The last paragraph of “Development of Radiation-induced Lesions:” section is now modified to address this comment. 
  4. This should not be an issue as BBB is broken by irradiation.

  1.  This is now addressed in the initial part of “Development of Radiation-induced Lesions” section., L28: The author needs to explain what is the “stem cell hypothesis”. There are potential several hypotheses.
  2. Here we are not talking about tissue specific stem cells. The author is referring to stem cells in general. For example stem cells such as mesenchymal stem cells that can differentiate into adipocytes, osteoblasts, myocytes, and chondrocytes in vivo and in vitro.
  3. I respect the reviewer’s comment but I do not see a need for diagrams.

Reviewer 2 Report

This paper reviews the state of art of stem cells and their potential in the treatment and regeneration of radiation induced normal tissue lesions. The paper is nicely written and I enjoyed reading it. The English is generally very good.

However, I miss some aspects, which should be added in Section 1 or in the discussion:

Cells can die after having been exposed to radiation with high enough energy. Many of them may also survive and suffer from unrepairable DNA damage. There is lot of knowledge about (DNA damage, DNA repair, double strand breaks and so on, tumor induction) available. It would be good to add some of this work as well. At least mention the mechanism.

In all experiments described in Section 6. - 12., high to very high doses of radiation were applied. I miss the information which doses are typical for therapeutic treatment and a critical discussion on the transferability of this results to lower doses. There is some short discussion on it in the conclusions, which I believe is not extensive and quantitative enough.

l.149: you cite a paper that investigates laser, I do not understand what lasers have to do with radiation. The energy level and mechanisms is not comparable with x-ray or particles.

Further, I miss the discussion on the radiation quality. Not all qualities have the same properties.

You name all type of stem cells but focus on the MSC. Please add some more (l144.) comment for this choice and briefly mention whether the other types are also investigated in this direction.

In the discussion, many aspects mentioned earlier are repeated. I suggest shortening it substantially. 

Minor changes

Spelling: there are a lot of double spaces. 

General: check the comma setting in the text

Some sentences sound weird, please look at it again: l. 18 While stem ..., l.40, l. 49 ( a sufficient ..), l.184 in fact, l.197 to to, l.241 ip ?, l.339 resulted, l.343 delete after, l.418 impair, the circulation, ..., l. 424 cause, l. 419 to , l. 426 radiation, l. 434 - 437, l. 512.

Author Response

Reviewer 2:

General comments:

While I understand the concern of the reviewer I would like to bring to his/her attention that this review, as the title implies, is on the therapeutic effects of MSCs. Therefore, discussing the DNA damage or lower doses of radiation is outside the scope of this review. However, in the second paragraph of “Discussions” it is pointed out that: “Not all radiation lesions are fatal. Radiation lesions heal after sublethal doses; when surviving cells in the irradiated region regenerate…….”

l.149: In this section the author talks about characteristic of the stem cells to point out that low level laser modifies their potential. He is not contrasting ionising radiation with lasers but simply pointing to the fact that low level laser can alter stem cells characterises exactly like if I say such chemical x alters stem cell’s morphology. 

radiation quality

This is a valid point and I added a sentence at the address this at the end of the first paragraph of Discussion.

Focus on the MSC: A sentence is added to the end of abstract to address this comment

With all respect I do not agree with shortening of discussions.

Minor changes

General: comma settings are reviewed

 l. 18 While stem ..., the sentence is now  modified.

Reviewer 3 Report

This is an interesting well-described study that explores therapeutic potentials of stem cells and extracellular vesicles in the treatment of radiation lesions. Authors argue that stem cell treatment is treatment of tissue deficits due to the inability of stem cells to replenish the lost cells in case of radiation lesions. Reason is that in case of severe damage due to irradiation, there is failure of endogenous stem cells to regenerate the damaged tissue. The contribution of exogenous stem cells makes it possible to correct this depletion in endogenous stem cells.

This study has the advantage of taking the history of stem cell treatment in irradiation. It also has the advantage of presenting the major works in this field. For these reasons this article fits perfectly in the special issue stem cells and. It addresses an interesting clinical area in an application of cell therapy of radiation lesions.

Minor revisions:

Page 2:

Add a reference, Sentence: “Radiation lesions are amenable to treatment by methods that result in repairing or regeneration of the damaged tissue. In fact stem cell transplantation in medical practice is not new and has been used for decades in bone marrow transplantation (ref missing).”

Page 3:

Sentences: “Recently, it was shown by many researchers including ourselves, that the effect of stem cells is exerted in a paracrine fashion (6-8). Transplanted stem cells, by integration with the host tissue, by mobilization of endogenous stem cells or by combination of both mechanisms, result in functional and structural improvements of injured tissues.”

Please explain with details, mechanisms of stem cell to repair irradiated tissue (paracrine).

You may add

When tissue homeostasis is disrupted, the supply of exogenous stem cells restores the balance between the disappearance of specialized cells and their renewal by endogenous stem cells (tissue homeostasis). However, exogenous stem cells rarely intervene by replacing tissue stem cells by trophic effect but rather by a trophic effect. This effect can be defined by the secretion of factors (paracrine effect) and or contact that prevents apoptosis of the tissue stem cells and also by the secretion of growth factors allowing tissue proliferation, angiogenesis, protection against oxidative stress and against an exacerbated inflammatory response.

Paragraph: Types of Stem Cells, please mentioned and explained the concept of pluripotency for iPS and ES cells, and multipotency for other cells.

For example:

Stem cells are classified by their potentiality into three main types; multipotent, pluripotent and totipotent. Totipotent stem cells donate an entire individual.

Pluripotency is the ability of certain cells to differentiate into the three embryonic layers (ectoderm, mesoderm and endoderm). The characteristic of ES and iPS cells is that they are multipotent cells.

Multipotency is the ability of stem cells to differentiate into one or two embryonic layers such as mesoderm and endoderm. In contrast, adult stem cells are multipotent cells.

Advantages and disadvantages of ES and IPS compared to adult stem cells must be explained.

Such as

ES and IPS cells have the advantage of indefinite renewal and the ability to differentiate into all cell types. This property gives them a role in replacing damaged cells by direct differentiation.

On the other hand, adult stem cells are limited in their proliferation and are difficult to isolate from adult or fetal tissues (e.g. umbilical cord). Adult stem cells can either differentiate to replace specialized cells but in a limited number of cases. This is the case, for example, with MSCs that differentiate into osteoblasts. On the other hand, when adult stem cells come to repairing tissue from which they did not originate, they preferentially act by trophic effect, such as MSCs to allow intestinal regeneration.

Page4:

Paragraph Homing of Transplanted Stem cells:

Can you explain and cite some of the factors responsible for cell homing?

For example

Domiciliation factors such as SDF-1 are known to allow the targeting of hematopoietic stem cells to the marrow when it needs to be recolonized by hematopoietic stem cells. The secretion of SDF-1 similarly allows the addressing of MSCs that express the CXCR4 molecule which is the receptor for the SDF-1 molecule. Many chemokines have been described as stem cell domiciliation factors.

Line “the hypothesis that transplanted stem cells migrate to tradiation-induced injury sites”: correct tradiation by radiation

Page 5:

Paragraph Studies on Hematopoietic System: first line, replace but In fact by but in fact

Page 6:

Paragraph Studies on Gut:

In order to highlight the work on the intestine, it would be interesting to detail the mechanisms involved specifically on the intestine.

For example. The effects of MSCs are a consequence of their ability to improve the renewal capability of the small intestine epithelium. MSC treatment favors the re- establishment of cellular homeostasis by both increasing endogenous proliferation processes and inhibiting radiation-induce apoptosis of the small intestine epithelial cells (Sémont et al. 2010, 2013). Furthermore MSC treatment decreased the interactions between Mast cells and nerve fibers and reversed mechanical visceral hypersensitivity (Durand C et al. Pain 2015).Proposed mechanisms are; MSCs release cytokines and growth factors such as, interleukin (IL)-11, human hepatocyte growth factor, fibroblast growth factor-2 and insulin-like growth factors. Each of these factor have been described earlier as facilitating intestinal mucosa repair, either through enhancement of cell proliferation or inhibition of epithelial cell apoptosis.

Gao Z, Zhang Q, Han Y, Cheng X, Lu Y, Fan L, Wu Z, Mesenchymal stromal cell-conditioned medium prevents radiation-induced small intestine injury in mice. Cytotherapy. 2012 Mar;14(3):267-73.

Durand C, Pezet S, Eutamène H, Demarquay C, Mathieu N, Moussa L, Daudin R, Holler V, Sabourin JC, Milliat F, François A, Theodorou V, Tamarat R, Benderitter M, Sémont A. Persistent visceral allodynia in rat exposed to colorectal irradiation is reversed by mesenchymal stromal cell treatment. Pain. 2015 Aug;156(8):1465-76.

Sémont A, Demarquay C, Bessout R, Durand C, Benderitter M, Mathieu N. Mesenchymal stem cell therapy stimulates endogenous host progenitor cells to improve colonic epithelial regeneration. PLoS One. 2013 Jul 29;8(7):e70170.

Sémont A, Mouiseddine M, François A, Demarquay C, Mathieu N, Chapel A, Saché A, Thierry D, Laloi P, Gourmelon P. Mesenchymal stem cells improve small intestinal integrity through regulation of endogenous epithelial cell homeostasis. Cell Death Differ. 2010 Jun;17(6):952-61

Page 10

Discussion: In order to summarize the scientific work on stem cells and irradiation, it would be useful to add a table summarizing the main studies organ by organ.

Page 13:

Conclusion: Given the great expertise in the field of the authors and their scientific contributions over the last two decades, it would be desirable to open the conclusion on new fields of research to be covered in the field of stem cells and irradiation

Author Response

Thank you very much for reading my article thoroughly and pointing out useful comments and suggestions. I appreciate this. I took into consideration all your comments except the following two pints:

part 1 because this concept is mentioned in different manner throughout the paper.

did not add a table summarizing the studies as the paper is already structured organ by organ.

Once again I thank you very much for your professional comments which surely improved the quality of my paper.

Round 2

Reviewer 1 Report

The author did not address my previous concerns.

Author Response

With thanks, I have already replied to your comments and no further action was needed on your second revision

Reviewer 2 Report

Unfortunately the author did not consider my advice to work on the improvement of the paper quality. I missed some detailed information on the radiation quality and dose. This was superficially answered. Also the discussion on cell damage mechanisms was considered not relevant. In this point the paper is too superficial. 

I also asked to shorten and sharpen the discussion. This was declined.

Altogether, only minor changes were done and the major issues were not considered relevant.  

Author Response

(The authors gave the same response as above.)
